# Application of a mobile laboratory using a Selected-Ion Flow-Tube Mass Spectrometer (SIFT-MS) for characterisation of volatile organic compounds and atmospheric trace gases

Rebecca L. Wagner[1], Naomi J. Farren[1], Jack Davison[1], Stuart Young[1], James R. Hopkins[1,2], Alastair C. Lewis[1,2], David C. Carslaw[1,3], and Marvin D. Shaw[1,2]

[1]Wolfson Atmospheric Chemistry Laboratories, University of York, York, YO10 5DD, United Kingdom
[2]National Centre for Atmospheric Science, University of York, York, YO10 5DD, United Kingdom
[3]Ricardo Energy & Environment, Harwell, Oxfordshire, OX11 0QR, United Kingdom

**Correspondence:** Marvin D. Shaw (marvin.shaw@york.ac.uk)

**Abstract.** Over the last two decades, the importance of emissions source types of atmospheric pollutants in urban areas has undergone significant change. In particular, there has been a considerable reduction in emissions associated with road vehicles. Understanding the role played by different source sectors is important if effective air pollution control is to be achieved. Current atmospheric measurements are made at fixed monitoring sites, most of which do not include the measurement of volatile organic compounds (VOCs) and so our understanding of the temporal and spatial variation of pollutants is limited. Here we describe the application of a mobile laboratory using a selected-ion flow tube mass spectrometer (SIFT-MS) and other trace gas instrumentation to provide on-road, high spatial and temporal resolution measurements of $CO_2$, $CH_4$, VOCs and other trace gases. We then present data illustrating the potential of this platform for developing source characterisation methods that account for the similarity in correlation between species. Finally, we consider the benefits of high spatial and temporal resolution measurements in characterising different types of sources, which would be difficult or impossible for single species studies.

## 1 Introduction

Air pollution in many urban areas is a major problem due to a myriad of emissions sources and dense populations leading to increased potential for human exposure. Among important air pollutants, volatile organic compounds (VOCs) are a class of pollutants that can significantly affect the chemistry of the atmosphere and human health. VOCs play an important role in atmospheric chemistry as they react rapidly with hydroxyl radicals and nitrogen oxides in the presence of sunlight to form products such as ozone ($O_3$) (Zhang et al., 2019) and peroxyacetyl nitrate (Roberts, 1990; Roberts et al., 2003). $O_3$ can cause respiratory irritation (Nuvolone et al., 2018) and damage to ecosystems (Grulke and Heath, 2020). Peroxyacetyl nitrate has been shown to have adverse effects on plant growth and human health at high concentrations (Vyskocil et al., 1998) and it has also been shown to thermally decompose to form $NO_x$, which leads to enhanced $O_3$ production (Heald et al., 2003). VOCs themselves can undergo gas-to-particle conversions to produce secondary organic aerosol (Kourtchev et al., 2016). Also, some

VOCs can cause acute irritations and damage to internal organs (Shuai et al., 2018) and chronic human exposure to benzene can induce haematological problems and cancer (Kampa and Castanas, 2008).

To combat air quality and health issues in urban areas, it is important that the composition and sources of VOCs are understood. Vehicular emissions of VOCs have historically been of central importance to issues such as $O_3$ and secondary organic aerosol formation. But the composition and sources of these species in urban areas is highly complex and it is difficult to determine the role played by VOCs from road vehicles relative to other sources. While VOC emissions from vehicle exhausts have been aggressively reduced over the past few decades in Europe through the introduction of technologies such as three-way catalysts on gasoline vehicles, they still present an important source of emissions, accounting for 4% of total UK VOC emissions in 2017 (Lewis et al., 2020).

Nevertheless, the reduction in vehicular emissions means it is likely that other sources of emissions, such as volatile chemical products (VCPs), solvent use, cooking, residential wood burning and industry, have become more important. Mass balance analysis by McDonald et al. (2018) concluded that emissions from VCPs now account for half of the fossil fuel VOC emissions in industrialised cities. Lewis et al. (2020) estimated that in the UK, VOC emissions from solvent use and industrial processes account for 63% of total VOC emissions. This change in the contribution of VOC emissions from different sources is not reflected in current VOC measurements made at stationary monitoring sites in the UK, which measure only 13 out of the 20 most significant VOCs (Lewis et al., 2020) due to set-up taking place 30 years ago, when road transport was the dominant VOC emissions source.

The current understanding of urban air pollution depends on hourly or daily measurements of a limited amount of atmospheric pollutants recorded at stationary monitoring sites. These measurements are unable to represent relative contributions of different source types and complex temporal and spatial variations of pollutants. To overcome this problem, mobile laboratories equipped with fast response instruments have been used for high spatial and temporal measurements of pollutants. Several studies have shown the use of on-road mobile laboratories for measurements of gaseous pollutants (Pirjola et al., 2004, 2014; Wu et al., 2013; Bush et al., 2015; Apte et al., 2017; Ars et al., 2020; Vojtisek-Lom et al., 2020) and particles (Pirjola et al., 2004, 2014; Bush et al., 2015; Saarikoski et al., 2017; Popovici et al., 2018; Alas et al., 2019). These studies highlight the use of mobile laboratories for the spatial mapping of pollutants and also other mobile measurements such as vehicle plume chase studies. However, these studies focus on measurements of common pollutants such as carbon dioxide ($CO_2$), nitrogen oxides ($NO_x$) and methane ($CH_4$), which typically represent specific emissions sources in urban areas, such as transportation or gas leakages.

VOCs in urban areas are emitted from a much wider range of sources than road transport, as discussed above. To make sure these complex sources are better understood, the number of compounds measured has to be expanded and their spatial emissions in urban areas better resolved. The incorporation of mass spectrometers into a mobile laboratory allows for measurements of dominant VOC species at high spatial resolution. Previous studies involving mobile measurements using mass spectrometry have used a Proton-Transfer-Reaction Mass Spectrometer (PTR-MS). A PTR-MS is a term used for an instrument which consists of an ion source that is directly connected to a drift tube and a mass analyzing system, which either consists of a quadrupole of time of flight mass analyser. Standard PTR-MS instruments are a Proton Transfer Reaction Quadrupole Mass

Spectrometer (PTR-QMS), which can detect and resolve product ion masses at single unit mass resolution. Airborne measurements of VOCs using PTR-MS have been carried out to identify dominant emissions sources (Shaw et al., 2015) and also to determine VOC fluxes (Karl et al., 2009; Vaughan et al., 2017). Early on-road measurements of VOCs using PTR-MS in a mobile laboratory were carried out by the Aerodyne Research Institute in vehicle plume chase experiments (Kolb et al., 2004; Herndon et al., 2005). More recent measurements carried out by Aerodyne (Knighton et al., 2012; Yacovitch et al., 2015), have focused on spatial mapping of petrochemical emissions. VOC emissions from the oil and gas industry, such as benzene, toluene and other aromatics have also been investigated using PTR-MS in a mobile laboratory (Warneke et al., 2014).

More recent studies by the NOAA group have used Proton Transfer Reaction Time-of-Flight Mass Spectrometers (further referred to as PTR-TOF) in mobile laboratories to measure VOCs in urban areas in the US. A PTR-TOF can detect and resolve product ions at much higher mass resolution with currently available commercial instruments having a mass resolution of greater than 4000. Measurements have revealed the emerging importance of varying emissions sources of VOCs- such as VCPs (Coggon et al., 2018; Gkatzelis et al., 2021a, b; Shah et al., 2020; Stockwell et al., 2020), which have been shown to be a major source of petrochemical emissions of VOCs in US cities and therefore play an important role in contributing to the formation of $O_3$ and secondary organic aerosol. Further measurements using the NOAA PTR-TOF have also shown substantial VOC emissions from concentrated animal feeding operations (Yuan et al., 2017) and from residential and crop residue burning (Coggon et al., 2016). Other measurements using PTR-TOF in a mobile laboratory to investigate emissions sources have been carried out to discriminate and spatially map VOC sources with the use of statistical methods (Richards et al., 2020) and also to investigate emissions from the oil and gas industry (Edie et al., 2020). These studies exhibit the potential of mass spectrometry in a mobile laboratory to distinguish and examine varying emissions sources of VOCs in urban areas.

The work described here differs from previous studies by using a selected-ion flow-tube mass spectrometer (SIFT-MS) to target 13 compounds at a 2.5 second time resolution, which has some important advantages when compared to PTR-TOF. An important advantage of the Voice200 ultra SIFT-MS (used in this study) is that it provides easy to use software and is suitable for a wide range of users, compared to PTR-TOF which requires considerable expertise. Therefore the method described in this paper could be used by non-research organisations, such as regulators or governments. Another difference is that SIFT-MS uses multiple reagent ions, which can be switched in real-time (discussed in Section 2.2.1). Some PTR-TOF instruments also utilise selective reagent ionisation, with the use of multiple reagent ions, but these have reagent ion switching times typically in the order of tens of seconds compared to SIFT-MS switching times of milliseconds. The use of multiple reagent ions also allows for measurement of a wider range of species, such as nitrogen dioxide ($NO_2$), nitrous acid (HONO) and ammonia ($NH_3$), which PTR-TOF is unable to measure, and separation of isomeric compounds (Lehnert et al., 2020). This means that mobile laboratories utilising SIFT-MS could be used for a wider range of applications.

Lehnert et al. (2020) concluded that SIFT-MS is sensitive enough to perform trace gas measurements in ambient air and it also performs well when analysing complex mixtures at varying humidities. A disadvantage of SIFT-MS is that it cannot measure as many compounds in the same time resolution as PTR-TOF, which can potentially measure hundreds of VOCs every second. But SIFT-MS can be used with careful selection of compounds, which ensures that target emissions sources can be investigated at an appropriate time resolution. The utilisation of multiple reagent ions allows for measurements of VOCs

and trace gases across a wide range of applications such as breath analysis (Španěl and Smith, 2008; Castada and Barringer, 2019), analysis of emissions from consumer products (Langford et al., 2019; Yeoman et al., 2020) and ambient air quality measurements (Prince et al., 2010; Crilley et al., 2019). These studies present successful measurements of a wide range of VOCs and atmospheric trace gases, showing that SIFT-MS is suitable for measurements in urban areas.

Here we will describe a mobile laboratory equipped with SIFT-MS and other trace gas instrumentation to provide high spatial and temporal resolution measurements of ($CO_2$), ($CH_4$), VOCs and other trace gases alongside meteorological and geospatial data. We also detail the steps carried out to assure data quality of these mobile measurements. Examples of data are shown illustrating that SIFT-MS is a suitable instrument for mobile measurements and highlight the potential of this platform for spatial mapping of pollutants. We also discuss source characterisation methods that account for the similarity in correlation between species and development of new methods that will provide further insight into emissions sources in urban areas.

## 2   Experimental

### 2.1   Description of the mobile laboratory

The platform used for mobile measurements is the WACL Air Sampling Platform (WASP), which is a Nissan NV400SE L3H2 with interior dimensions of length 3450 mm, width 1650 mm, height 1750 mm and a payload of 1000 kg. The walls and floor in the rear of the van are overlaid and are fitted with 50 mm of insulation. The rear of the van contains an air conditioning system, which can be controlled from the driver cab and maintains the rear of the van at a constant temperature, overcoming any instrumental heating. The instruments are mounted and secured with aircraft-style L-tracks, which are built into the flooring and the ceiling. These allow for ratchet strapping of the SIFT-MS instrument and the instrument rack, where the computer system and the ultra-portable greenhouse gas analyser (UGGA) sit. The power in the van is supplied by two 12VDC 230Ah batteries that are charged whilst driving by a 240VAC inverter. Alternatively they can be charged when stationary by an external mains power port. The total battery life with the instruments and air conditioning running is about 2 hours whilst driving.

The front-facing sample inlet of the van sits at 2.25 m off the ground, at windscreen height of the WASP, and is made from a 6 m length of PTFE tubing with a diameter of 12.5 mm (1/2 inch). Sample air is drawn from the outside through the sample inlet by a pump located in the rear at a flow rate of 40 SLPM, and is then fed into the instruments mounted in the rear. Real-time location of the WASP is recorded by a Garmin GPS 18x PC and a measurement of wind speed and direction is measured using a Gill 2D Ultrasonic Wind Sensor (which was not used in this study), both of which are fitted on the roof of the WASP at a height of 2.5 m. Data outputs of the roof sensors and the UGGA are stored on a computer system in the van and the outputs of the SIFT-MS are stored in the internal system of the instrument. Ethernet port connections between the computer system and the SIFT-MS mean that data for all of the instruments can be visualised in real-time in the driver cab whilst measurements are carried out. A schematic of the WASP is shown in Figure 1 and further details of the instrumentation are provided in Table 1. Note that the instrument fit in the WASP is flexible and is easily altered due to the relatively large size and payload that the WASP can accommodate.

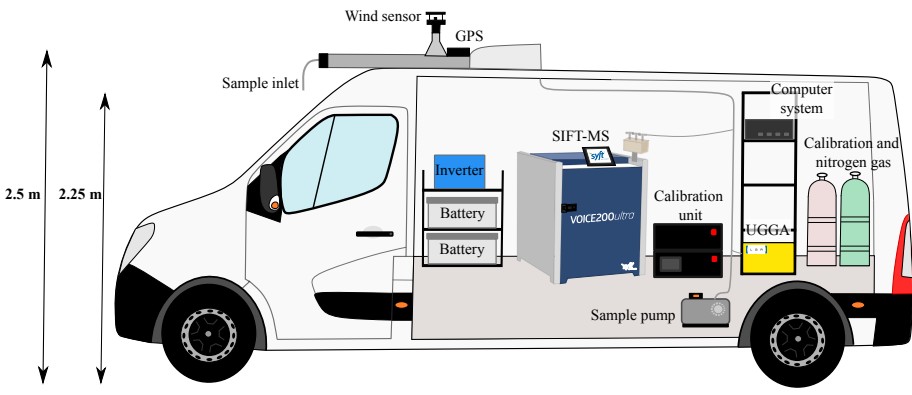

**Figure 1.** A graphic of the WACL Air Sampling Platform (WASP) used for mobile measurements.

| Instrument | Time resolution (s) | Compounds | Power @ 240V (W) |
| --- | --- | --- | --- |
| **Voice200 ultra SIFT-MS** | 2.5 | 13 organic and inorganic gases | 1100 |
| **Los Gatos Research UGGA** | 1 | $CH_4$, $CO_2$, $H_2O$ | 340 |
| **Gill 2D Ultrasonic Wind Sensor** | 1 | Wind speed and direction | 120 |
| **Garmin GPS 18x PC** | 1 | Vehicle speed, direction and location | 70 |

**Table 1.** Details of the WASP instrumentation.

## 2.2 Instrumentation

### 2.2.1 Selected-Ion Flow Tube Mass Spectrometer (SIFT-MS)

A Voice200 ultra SIFT-MS manufactured by Syft Technologies (Christchurch, New Zealand), was used to quantify VOCs and inorganic gases. The SIFT-MS principles of operation are discussed in detail elsewhere (Smith and Spanel, 1996; Smith and Španěl, 2005), but a brief outline is included here. The instrument consists of a switchable reagent ion source capable of rapidly switching between multiple reagent ions: $H_3O^+$, $NO^+$ and $O_2^+$, which are generated in a microwave plasma ion source, from a mixture of air and water at a pressure of approximately 440 mTorr. The reagent ions are then extracted into the upstream quadrupole chamber maintained at a pressure of approximately $5 \times 10^{-4}$ Torr, using a $70\,\mathrm{L\,s^{-1}}$ turbo-molecular pump, and then pass through an array of electrostatic lenses and the upstream quadrupole mass filter. Those not rejected by the mass filter are injected into the flow tube where they are thermalized in a stream of nitrogen prior to selectively ionising target analytes. The product ions then flow into the downstream quadrupole mass filter and the secondary electron multiplier detector, where the ions are separated by their mass-to-charge ratios (m/z) and the ion counts are measured. The mixing ratios of analyte compounds in the flow tube are calculated using the ion-molecule reactions that take place within the SIFT-MS using Equation

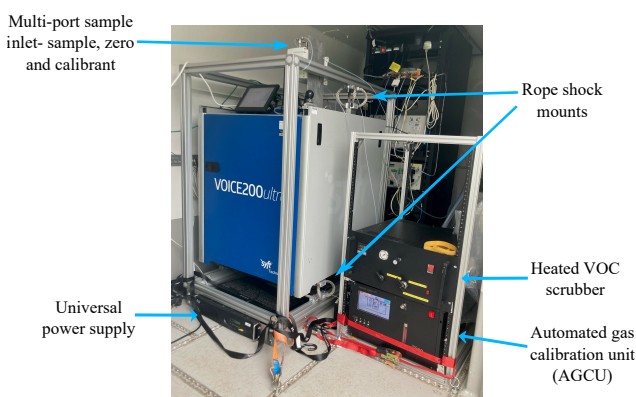

**Figure 2.** The Voice200 ultra SIFT-MS with in-house built multi-port inlet and heated VOC scrubber, alongside the custom built automated gas calibration unit (AGCU).

1, where $[A]$ is the analyte mixing ratio, $\gamma$ is the instrument calibration factor, $[P^+]$ is the product ion, $[R^+]$ is the reagent ion, $t_r$ is the reaction time and $k$ is the rate constant.

$$[A] = \gamma \times \frac{[P^+]}{[R^+]t_r k} \tag{1}$$

Figure 2 shows the SIFT-MS inside the WASP, with an in-house multi-port sample inlet capable of autonomously selecting between sample, zero and calibrant gases using the instrument software (Labsyft 1.6). The multi-port inlet uses 3 PTFE internally coated solenoid valves (12VDC, Gems). The SIFT-MS is suspended in a rope-shock mounted rack, which reduces 3-dimensional vibration the instrument is subjected to during mobile measurements. Whilst driving, the SIFT-MS was operated using a flow tube pressure of 460 mTorr, a nitrogen carrier gas (Research grade, BOC) with a flow rate of 0.6 Torr $\mathrm{L\,s^{-1}}$ and

a sample flow rate of 100 SCCM. On the right of Figure 2 is a custom built automated gas calibration unit (AGCU) and a heated VOC zero air generator. The heated VOC scrubber consists of palladium-coated alumina pellets heated to 380°C which produces zero air whilst maintaining the humidity of the sample gas. Mass flow controllers (MFCs) (Alicat) in the AGCU measure and control the flows of diluent zero air and the VOC standard (1ppm certified National Physics Laboratory, UK), and allow for controlled dilution ratios in the ppt-ppm range. Automated step-wise changes to the dilution ratios are made,

which generates a multi-point calibration curve for routine external calibration of the SIFT-MS. Figure 3 shows a simplified schematic of the internal gas flow paths in the AGCU and calibration of the compounds is discussed further in Section 3.1.

     The SIFT-MS was used to measure 13 different VOCs and inorganic gases during mobile measurements around the city of York, UK. The compounds targeted with the SIFT-MS were chosen to cover a range of emissions sources to help with source apportionment analysis. The compounds measured by the SIFT-MS and the corresponding reagent ions, molecular masses

and product ion chemical formulas are shown in Table 2. To maximise spatial data density during mobile measurements, the instrument acquisition rate was minimised with only a single product ion monitored for each compound. Therefore the sampling method used during measurements has an acquisition rate of 2.5 seconds with a 90 ms ion dwell time.

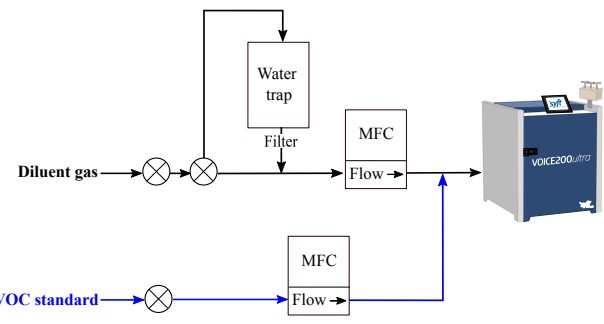

**Figure 3.** Schematic of the internal gas flow paths in the automated gas calibration unit (AGCU).

| Reagent Ion | Compound | MM[1] | Product Ion |
|---|---|---|---|
| $O_2^+$ | $NO_2$ | 46 | $NO_2^+$ |
| $NO^+$ | Isoprene/Furan | 68 | $C_5H_8^+$ |
| | Benzene | 78 | $C_6H_6^+$ |
| | Acetone | 88 | $(CH_3)_2NO^+$ |
| | Toluene | 92 | $C_6H_5CH_3^+$ |
| | m-Xylene ($C_2$-alkyl benzenes) | 106 | $C_6H_4(CH_3)_2^+$ |
| | 1,2,4-Trimethylbenzene ($C_3$-alkyl benzenes) | 120 | $C_6H_3(CH_3)_3^+$ |
| | Total Monoterpenes | 136 | $C_{10}H_{16}^+$ |
| $H_3O^+$ | Methanol | 33 | $CH_3OH^+$ |
| | Acetaldehyde | 45 | $CH_3CHO^+$ |
| | Ethanol | 47 | $C_2H_5OH^+$ |
| | HONO | 48 | $HNO_2^+$ |
| | 1,3-Butadiene | 54 | $(C_2H_3)_2^+$ |

[1]Molar Mass in $\mathrm{g\,mol^{-1}}$

**Table 2.** The compounds measured by the SIFT-MS and their corresponding reagent ions, molecular masses and product ion chemical formulas.

### 2.2.2 Ultra-portable Greenhouse Gas Analyser (UGGA)

A Los Gatos Research Ultra-Portable Greenhouse Gas Analyser (UGGA) was used to quantify carbon dioxide ($CO_2$), methane
($CH_4$) and water vapour ($H_2O$). The UGGA instrument uses off-axis integrated-cavity output spectroscopy (off-axis ICOS) to
quantify mixing ratios of gaseous species, which has been described in detail previously (Gupta, 2012), but a brief description
is included here. Off-axis ICOS uses a laser and an optical cavity in an off-axis configuration (Tan and Long, 2010), which

enhances the measured absorption of light by a sample by creating an effective optical path length of several thousands of meters. The measured absorption spectra is recorded and when this is combined with the measured gas temperature and pressure in the cell, effective path length and known line strength, it can be used to determine a quantitative measurement of mixing ratio.

The UGGA was used at a 1 Hz time resolution, with a response time of 10 seconds. The precision of the UGGA over 1 second is 2 ppb for $CH_4$ and 500 ppb for $CO_2$ and the limit of detection (LoD) is 3 ppb for $CH_4$ and 800 ppb for $CO_2$. The precision and LoD of the UGGA was calculated using $2\sigma$ and $3\sigma$, respectively, of gas canister only measurements made whilst driving. The measurement range of the UGGA is 0.01-100 ppm and 1-20,000 ppm for $CH_4$ and $CO_2$ respectively. A quantitative measurement of mixing ratio could be determined directly from the UGGA without the need for calibration during each drive, but the UGGA was calibrated with external gas cylinders before and after the measurement period.

## 2.3 Measurement location

In the summer of 2020, the WASP was used to make measurements around the city of York, UK. Figure 4 shows the measurement route that was driven by the WASP. The route starts at the University of York and then passes into and around the inner ring road of the city and has a total distance of 15.1 kilometers. York has a population of approximately 200,000 people and air pollution in the city is thought to be dominated by vehicle emissions, especially around the inner ring road due to congestion. The measurement route in York was designed to capture a variety of potential emissions sources, such as hairdressers, beauty salons, dry cleaners and eateries, to determine dominant sources. Measurements were carried out for a total of 10 days, between the 30th June 2020 and the 23rd July 2020, during periods of dry weather between the hours of 10:00 and 17:00. The route was driven 30 times in total and the dates and times of each drive are included in the appendix (Table A1). It should be noted that during the measurement period the Covid-19 stringency index was 64.35 (taken from: COV), indicating reduced economic and traffic activity. This could affect both the concentration and detection of different VOC species due to possible decreased emissions.

## 3 Results

### 3.1 Calibration and data quality assurance

The mixing ratios of all of the compounds measured by the SIFT-MS were dependent on the compound specific rate constant and daily generated instrument calibration factor (ICF ($\gamma$), Equation 1). The daily ICF was derived using a 2 ppm gas standard to validate the mass dependent ion transmission of the instrument, details of the gas standard are included in the appendix in Table B1. A zero offset was applied for all of the compounds which was taken from zero air for VOCs and from a nitrogen blank for other compounds, such as HONO and $NO_2$. In the case of acetaldehyde, total monoterpenes, $NO_2$ and HONO, the mixing ratios were determined directly from the daily ICF. The remaining compounds were externally calibrated using the automated gas calibration unit (AGCU, discussed in Section 2.2.1) and a 1 ppm 14-component VOC gas standard (National

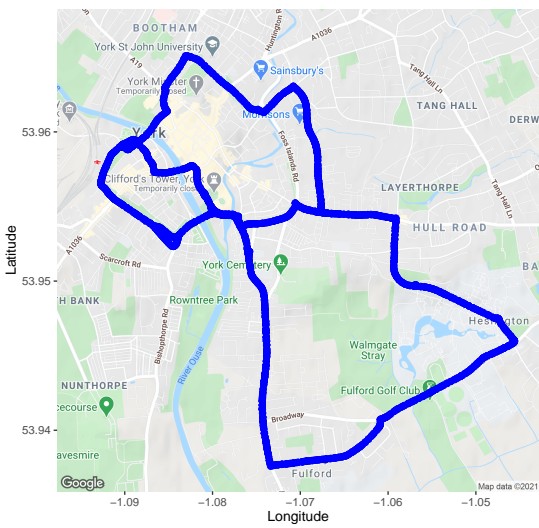

**Figure 4.** Measurement route driven by the WASP for mobile measurements around York (© Google). Visualised by the *ggmap* R package (Kahle and Wickham, 2013).

Physical Laboratory). These compounds include: acetone, benzene, butadiene, ethanol, isoprene, methanol, m-xylene ($C_2$-alkyl
benzenes), toluene and trimethlybenzene ($C_3$-alkyl benzenes). Figure 5 shows the multi-point calibration of these compounds
which was carried out at the start and end of each day (pre- and post-drive). Both pre- and post-drive SIFT-MS calibrations
were performed to ensure specific VOC compound sensitivities did not change during mobile operation. Due to operational
time constraints associated with switching on the SIFT-MS at the start and end of each day, the chosen calibration procedure
had to be as concise as possible. In order to assess daily both intra- and inter-calibration reproducibility, duplicate calibration
steps were carried out both before and after mobile operation. The calibrations were performed at VOC mixing ratios of 10, 5,
1 and 0 ppb and each of the mixing ratio steps lasted for 3 minutes.

The left- and right-hand steps of the calibrations showed good agreement (as shown in Figure 5). The left-hand calibration
steps were used to condition the internal surfaces and SIFT-MS inlet and therefore were not used for calibration. Compound
specific calibration curves were obtained from the average of the last minute of each 3 minute right-hand steps of the post-drive
calibrations and data influenced by mixing ratio transitions was removed. Calibrations were carried out daily and results were
applied to the mixing ratio data obtained during mobile measurements. Figure 5 also shows good agreement for the majority of
the compounds between the pre- and post-drive calibrations and differences in the mixing ratios may have been due to changes
in ambient temperature, as pre-drive calibrations were usually performed in the morning when the temperature was lower.
This may have lead to the alcohols sticking on the sample lines or mixing with condensation in the line during calibrations.
Additional data from the calibrations, including the coefficient of determination ($R^2$) values and ion counts per second per ppb
(cps ppb $^{-1}$) are included in Table 3.

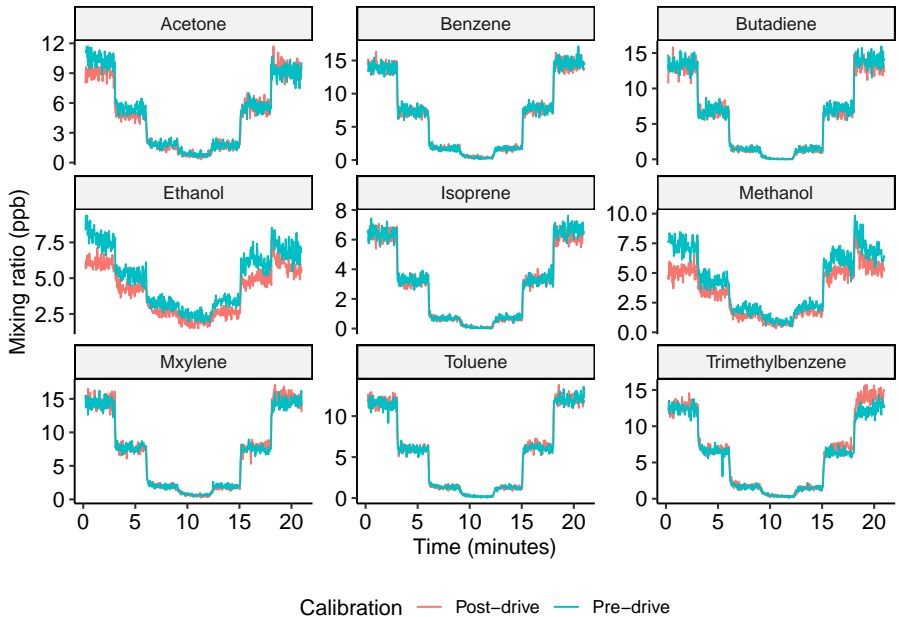

**Figure 5.** Multi-point calibration for VOC gas standard compounds over 10, 5, 1 and 0 ppb. The blue line shows the calibrations performed pre-drive and the pink line shows the calibration performed post-drive.

Table 3 shows the precision and the limit of detection (LoD) for the SIFT-MS compounds, which were calculated using 2 (precision) and 3 (LoD) times the standard deviation ($\sigma$) of measurements made when sampling zero air or nitrogen gas. The precision and the LoD shown in Table 3 were calculated over 2.5 seconds, which is the acquisition rate of the SIFT-MS 215 measurements.

To ensure that the VOC mixing ratios measured by the SIFT-MS were independent of instrumental noise, instrument dark counts using the $H_3O^+$ reagent ion, (m/z 25) were routinely measured during mobile operation, which is shown in the appendix (Figure A1) along with benzene concentration and counts per second (cps) for the benzene product ion (m/z 78). There was typically minimal instrument noise (0-40 cps for m/z 25) observed during a 45 minute mobile measurement period. Furthermore, 220 significant increases in the mixing ratio of benzene corresponded with large increases in m/z 78 (> 40 cps), therefore showing that these were due to real increases in ambient concentrations. Periods of elevated noise (>100 cps) were routinely removed to improve measurement accuracy. During the 10 days of measurements, a total of approximately 26.5 hours of measurements were made and only 3 minutes of the measurements had to be removed due to excess instrument noise, which may have been due to extreme vibrations or movement when driving. Movement and vibration effects on instrument noise whilst driving were 225 further investigated by sampling the SIFT-MS instrument on nitrogen only, which is shown in the appendix (Figure B1). The run did not show any significant changes in instrument noise whilst driving and there were no increases in compound mixing ratios, therefore showing that the driving motion had only a minimal effect on measurement accuracy.

| Species | Calibration data | | LoD and precision | |
| --- | --- | --- | --- | --- |
| | $R^2$ | cps ppb$^{-1}$ | LoD | Precision |
| Acetone | 0.98 | 133 | 0.22 | 0.15 |
| Benzene | 0.99 | 215 | 0.20 | 0.13 |
| 1,3-Butadiene | 0.99 | 149 | 0.06 | 0.04 |
| Ethanol | 0.88 | 232 | 0.27 | 0.18 |
| Isoprene | 0.99 | 205 | 0.04 | 0.02 |
| Methanol | 0.81 | 358 | 0.17 | 0.11 |
| m-Xylene ($C_2$-alkyl benzenes) | 0.99 | 473 | 0.33 | 0.22 |
| Toluene | 0.99 | 463 | 0.11 | 0.08 |
| 1,2,4-Trimethylbenzene ($C_3$-alkyl benzenes) | 0.99 | 449 | 0.14 | 0.09 |
| Acetaldehyde | - | - | 0.41 | 0.27 |
| Monoterpenes | - | - | 0.11 | 0.07 |
| $NO_2$ | - | - | 2.48 | 1.65 |
| HONO | - | - | 0.31 | 0.21 |

**Table 3.** The coefficient of determination ($R^2$) and the ion counts per second per ppb (cps ppb$^{-1}$) calculated from the SIFT-MS calibrations. The limit of detection (LOD) and precision of compounds measured by the SIFT-MS (in ppb) for every 2.5 second measurement. Compounds in the upper part of the table were externally calibrated.

### 3.2 Summary of measurements and spatial distribution

Figure 6 shows a statistical summary of the mobile measurements carried out in York in a box and whisker plot. The plot shows that butadiene, isoprene, HONO and monoterpenes mixing were consistently below 0.5 ppb, suggesting a lack of emission sources for these compounds in York, either generally or as a result of Covid-19 restrictions. The majority of the measurements of butadiene and HONO were below the limit of detection, so these compounds are not to be included in any further analysis. For the other compounds there is greater variation in the mixing ratios. The greatest variation for compounds measured by SIFT-MS is seen for vehicle-emissions related compounds, such as benzene, toluene, m-xylene and $NO_2$, in addition to other species such as acetaldehyde and ethanol. The variation in ethanol may also be related to emissions from vehicles or fuel evaporation due to ethanol addition in gasoline fuel. Ethanol variation could also result from use of VCPs. Acetaldehyde variation may be due to emissions from vehicles as an oxidation product or other atmospheric processes. It is difficult to say exactly what emissions source is responsible for the variation in ethanol and acetaldehyde, but methods such as spatial mapping, correlation and ratio analysis can be used to investigate different sources. It is still worth noting that the variation of all of the compounds is still small, which is likely due to Covid-19 restrictions and reduced economic and traffic activities around York. The majority of

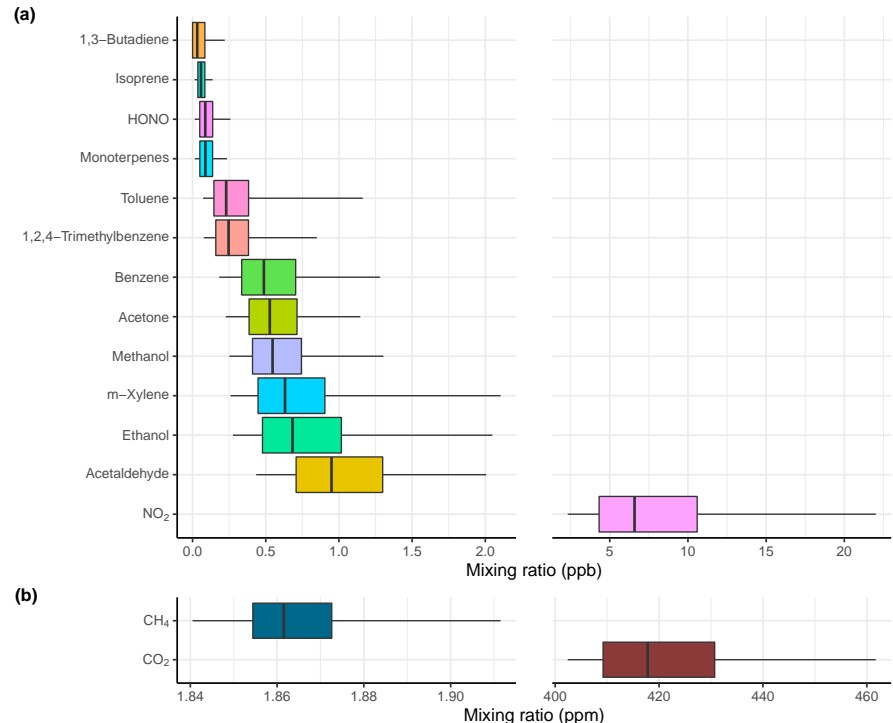

**Figure 6.** Summary of measurements made by a) the SIFT-MS (in ppb) and b) the UGGA (in ppm) during 30 repeat drives around York. The box outline contains the 25th to the 75th percentile and the middle line shows the median mixing ratio for each compound. The whiskers represent the 5th and 95th percentile for the mixing ratios of each compound. It is worth noting that $NO_2$ and $CO_2$ are on individual scales due to differing ranges in their mixing ratios.

the measurements (90.2 % - excluding butadiene and HONO) made by the SIFT-MS significantly exceeded the LoD calculated for each compound (Table 3) and the 30 metre aggregated median concentrations used for spatial mapping were on average a factor of 2.2 times higher than the LoD. This provides confidence that the observed compound peaks correspond to real increases in ambient concentrations. The typical spatial distance for the 2.5 seconds recorded by the SIFT-MS was 14 metres

as the average speed of the mobile laboratory was 20 kilometers per hour during measurements around York.

To spatially map the measurements made by the WASP mobile laboratory the data points were "snapped" to the nearest 30 metre segment of road using GPS data (Apte et al., 2017). The points in each 30 metre segment of road were then aggregated to give the median value. The median was selected as it would represent a realistic picture of air pollution and emissions sources on the measurement route and remove any biases that may occur from directly sampling vehicle exhausts.

Figure 7 shows the mixing ratios of benzene, toluene, ethanol and $NO_2$ recorded along the York measurement route, as examples of spatial mapping of compounds using the WASP and the SIFT-MS. The plot shows low mixing ratios on Heslington Lane (blue/green at the bottom of Figure 7) for all of the compounds. The mixing ratios increase around the city centre, where there is higher congestion and emissions may be dominated by vehicles. $NO_2$ mixing ratios considerably increase in the city

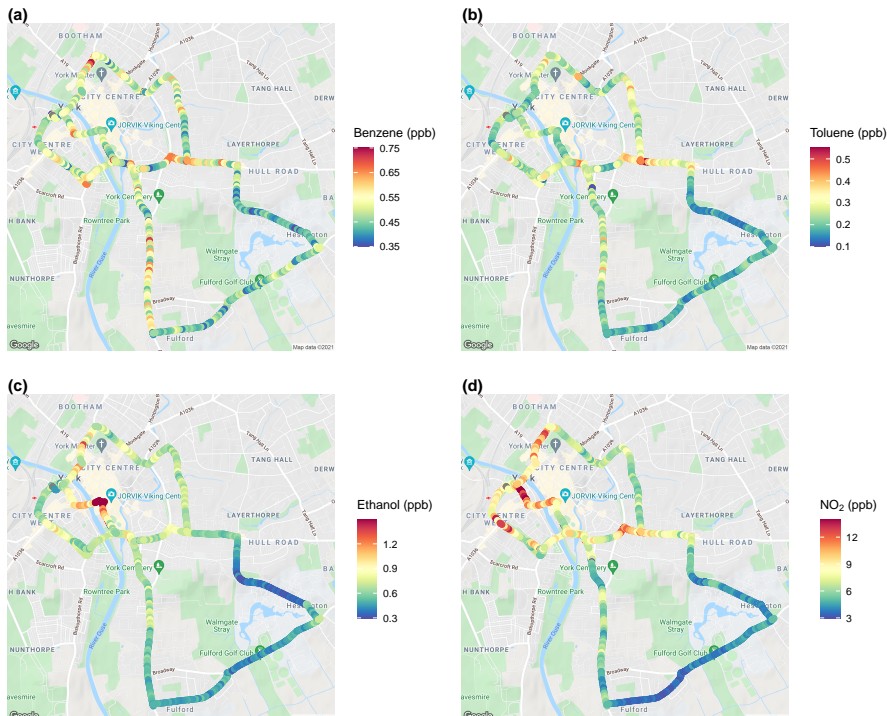

**Figure 7.** Spatial mapping of median values of (a) Benzene (ppb), (b) Toluene (ppb), (c) Ethanol (ppb) and (d) $NO_2$ (ppb) from mobile measurements around York (© Google). Visualised by the *ggmap* R package (Kahle and Wickham, 2013).

centre to above 12 ppb and elevated levels are seen at road junctions, which is expected as $NO_2$ is strongly indicative of diesel
vehicles. Particularly high levels of $NO_2$ are observed past the train station and on roads with many bus stations. Increases in
the median benzene and toluene mixing ratios are relatively small, but they show good similarities to each other with some
elevated levels at road junctions, which is indicative of emissions from road vehicles. Ethanol is at background levels for the
majority of the route, with considerable increases seen in the city centre of above 1.2 ppb (red in the centre of the map), which
may possibly originate from businesses in the city centre, such as bakeries and breweries. It is important to look at correlations
of species with one another in order to determine sources of VOC compounds in York.

### 3.3  Spatial correlation mapping

Spatial mapping is useful as it can highlight hotspots of VOCs and trace gases, but they are insufficient in determining and
separating emissions sources in urban areas. A method for source apportionment is to consider the correlation between many
species over particular roads or areas that measurements have been made. Areas that contain different emissions sources
should have different correlations, for example an area dominated by vehicle emissions should show strong correlations with
VOCs associated with vehicles such as benzene and toluene and these correlations will be different to other sources, such as

evaporative emission of solvents. Figure 8 shows the Spearman correlation of compounds for all of the measurements made by the SIFT-MS and UGGA in York.

Hierarchical clustering is applied to the correlation matrices to group species that are most similar to one another. For example, benzene, toluene, $C_2$-alkyl benzenes and $C_3$- alkyl benzenes appear next to each other in Figure 8 and show a clear cluster, indicating similar behaviour, likely related to their common sources of gasoline evaporation and engine exhaust emissions. There are other clusters of VOCs which are indicative of different source types - for example commons solvents such as acetaldehyde, methanol and acetone are clustered together in linked urban emissions likely not relating to road transport. Other trace gases associated with tailpipe emissions also appear in a single cluster, e.g $NO_2$ and $CO_2$. Ethanol is equally correlated to the vehicle-related cluster (benzene, toluene, $C_2$-alkyl benzenes and $C_3$- alkyl benzenes) and a non-vehicle related cluster (acetaldehyde, methanol and acetone), indicating that the sources of ethanol in York are complex and that there is not one dominating emissions source. In the future, the aim is to apply this method to smaller spatial scales, where it is expected that spatially varying patterns of correlation may enable specific emissions sources to be identified, potentially at the individual building scale.

The correlation plot shown in Figure 8 is also useful as it can be used as a guide for further analysis. Correlations between species can be investigated on a smaller scale through ratio analysis using quantile regression, such as the toluene to benzene ratio, which will be discussed in Section 3.4.

## 3.4 Evaporative source characterisation

Determining the importance of different sources in the emissions of atmospheric pollutants is difficult and calculating the ratios between compounds can be used as an indicator of dominating emissions sources. The toluene to benzene (T/B) ratio has been used in many studies to determine the dominant source of these compounds and it will be used here. Both toluene and benzene are emitted directly from vehicle exhausts and more widely by industry and may also be present due to evaporative emissions from fuels and solvents. Therefore, the T/B ratio is a useful indicator of the relative importance of different sources. A T/B ratio of 1-2 indicates a traffic influence (Langford et al., 2019; Simpson et al., 2020) and a T/B ratio of over 2 indicates solvent or evaporative influences (Simpson et al., 2020).

A characteristic of mobile measurements is the transient nature of emission sources. For sources such as road vehicle exhaust, the measurements will depend on the traffic conditions, which are inherently variable. With repeat measurements, these variations can be averaged-out to reveal consistent spatial patterns (Apte et al., 2017), which is shown by the spatial maps, in Figure 7. Some sources can, however, be highly transient such as the evaporative emissions from road vehicle fuel stations. The emission source from fuel stations can be highly variable and can depend on many variables, for example, the number of vehicles being refuelled. Furthermore, these sources will be intermittently detected due to the prevailing wind direction. It will often be the case that no plume is detected because the wind is from the wrong direction.

In terms of concentration measurements, some source contributions may show as infrequent high concentrations (or ratios of concentrations) that have little or no effect on median values. Taking the median value of concentrations, as used by Apte et al. (2017) and in Figure 7, will down-weight the contribution from less frequent, higher concentrations. Considering the median

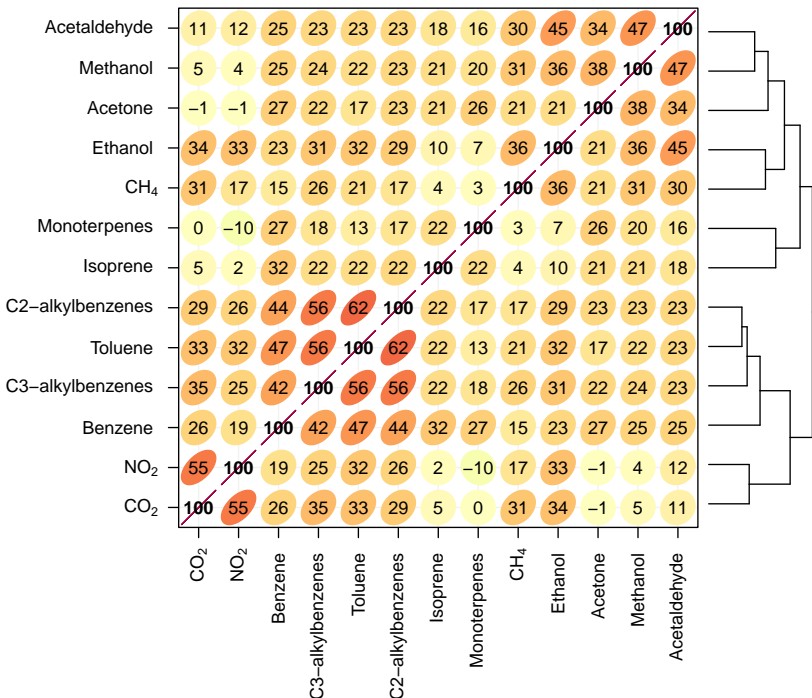

**Figure 8.** Spearman correlation of the compounds measured using the WASP and the SIFT-MS during measurements in York. Note that hierarchical clustering is applied to the correlation matrices to group species that are most similar to one another. A higher correlation coefficient between species is represented by a higher number, a darker red colour and an ellipses (shape). The lines on the right-hand side show the hierarchical clustering between compounds and represent clusters of species with similar patterns/behaviours.

is useful when investigating consistent emissions sources, however, there is potentially important information and intermittent sources that can be missed.

As an example, we consider T/B ratio as being indicative of both road vehicle exhaust and potentially evaporative emissions. The T/B ratio can be derived from the slope of a ordinary least squares (OLS) regression relating the two species. The linear regression ratio between toluene and benzene derived across all of the repeat drives, per a 30 metre segment of road, is shown plotted on a map of York in Figure 9. Figure 9 shows hotspots in the T/B ratio of above 4 on Hull Road and James Street (which joins Hull Road, both of which are labelled). On Hull Road there are two road vehicle fuel stations and on James Street there is an industrial area including a bus depot. The increased T/B ratio along these roads is significant as it is continuously high for multiple 30 metre segments. These two roads only account for about 5% of the route and may represent an important source of toluene. However, OLS regression considers the mean response and does not capture the full distribution of relationships that exist between toluene and benzene across all 30 of the repeat drives. To address this issue we adopt quantile regression to provide increased distributional information.

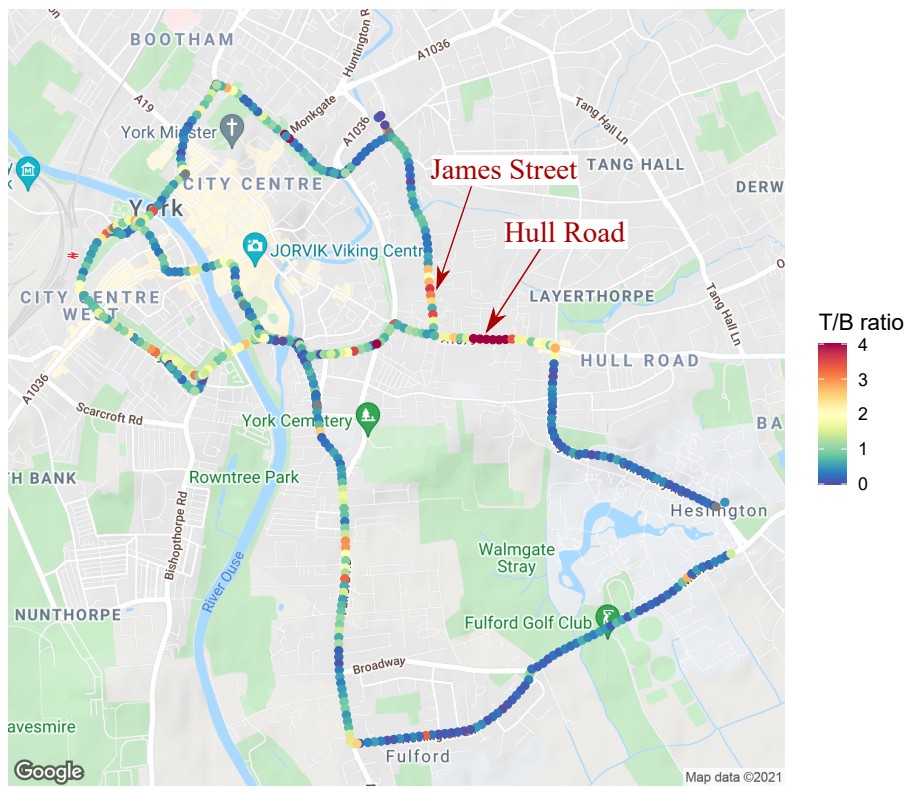

**Figure 9.** A spatial map showing the toluene to benzene (T/B) ratio calculated using ordinary least squares (OLS) regression (© Google). The ratio was calculated using toluene and benzene measurements made during all 30 repeat drives around York and each point represents the ratio for a 30 metre segment of road. Elevated ratios on Hull Road and James Street are labelled with red arrows. Visualised by the *ggmap* R package (Kahle and Wickham, 2013).

While an OLS regression line minimises the distance of the trend line to each point of data, quantile regression attempts to define a line such that a certain proportion of the data is found above and below it (Koenker, 2021). For example, the median quantile regression slope would have 50% of the data either side of it, whereas the slope associated with the 90th percentile would have 90% of the data below and 10% above. Through a quantile regression approach, the influence on the response variable by the predictor variable can be understood at different values of the response variable. The value and/or significance of a slope estimate could significantly vary between different quantiles. For example, a predictor variable may have a large influence on the response variable associated with "average" members of a population, but not those belonging to lower or higher quantiles. In the context of mobile measurements, high quantile values are used to explore the nature of transient evaporative sources that can be difficult to identify through OLS regression.

In a further development, rather than dividing the road network into discrete, non-overlapping 30 segments to calculate the T/B ratio, a Gaussian kernel smoother is used. This approach has the advantage of providing a continuous weighting function that gives more weight to data close to the location of interest, while down-weighting data collected further away. The approach

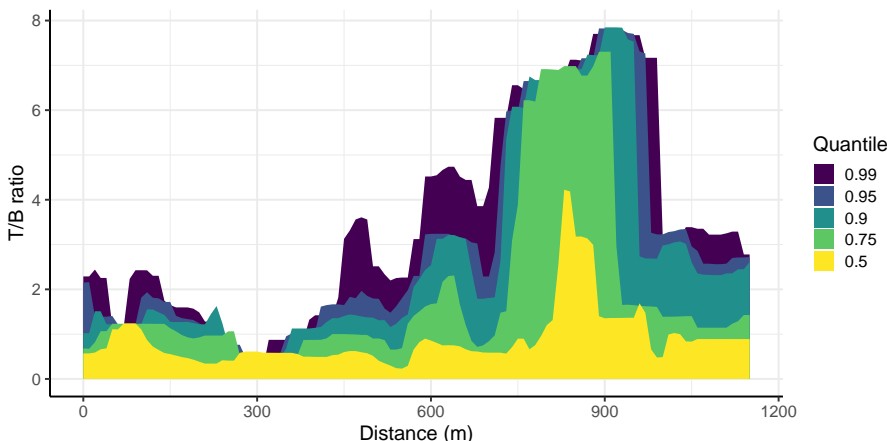

**Figure 10.** Toluene-to-benzene (T/B) ratio along the distance of Hull Road at quantiles of 0.99 (purple), 0.95 (dark blue), 0.9 (turquoise), 0.75 (green) and 0.5 (yellow) calculated using a Gaussian kernel smoother. The large peak at around 860 metres corresponds to a road vehicle fuel station, indicating evaporative emissions from this source.

also avoids arbitrarily dividing the road up into sections, where it can be difficult to determine an appropriate section length. Additionally, the Gaussian kernel weighting can be used directly in quantile regression modelling (or other statistical models), effectively providing a continuous estimate of the T/B ratio spatially, for different quantile values.

Figure 10 shows the results of the Gaussian kernel smoother along the distance of Hull Road at varying quantiles of 0.5, 0.75, 0.9, 0.95 and 0.99, using benzene and toluene measurements from all of the repeat drives. The plot shows that as the 330 percentiles increase, so does the T/B ratio. This is significant as the higher quantiles represent intermittent sources that would be missed by considering only the 50th percentile. While the median T/B ratio peak is around 4, the higher quantiles (0.75 to 0.99) converge on a ratio closer to 7 or 8, which will be more representative of the intermittent evaporative source than the median slope value or the OLS regression slope. On Hull Road there is a road vehicle fuel station situated at around 860 m, which corresponds to the large increase in the T/B ratio and this is highly likely due to evaporative emissions from the fuel 335 station, due to the absence of other potential sources at this location.

Figure 11 shows a comparison of the T/B ratio for the roads with a higher T/B ratio (Hull Road and James Street) — labelled as evaporative — and the remainder of the route. Figure 11 shows that at higher quantiles, the evaporative area has a much higher T/B ratio than the remainder of the route (6.70 compared to 4.12), but that it is still significant and corresponds to an intermittent source. The figure also shows that at a lower quantiles of 0.5 and 0.75, both of the comparison plots agree well, 340 with similar T/B ratios, representing emissions from vehicle exhausts. These results suggest that on the whole route, there is only a minor contribution of evaporative sources (about 5% of the length) that can clearly be identified in the data. Extending the route in York or making measurements in other urban areas with a wider range of source types would help develop these methods further.

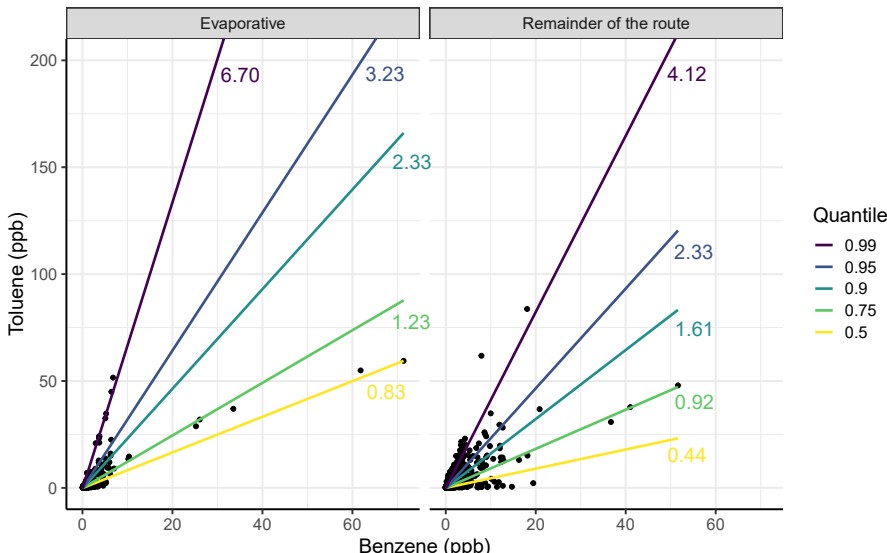

**Figure 11.** The toluene to benzene (T/B) ratio for the evaporative areas (Hull Road and James Street) and for the remainder of the route at quantiles of 0.99 (purple), 0.95 (dark blue), 0.9 (turquoise), 0.75 (green) and 0.5 (yellow), with the corresponding slope values also displayed. The toluene and benzene values for each of the comparative figures were produced using data collected from all 30 repeat measurement drives.

## 4    Conclusions and future applications

The SIFT-MS has been used to perform high temporal and spatial measurements of multiple VOCs in an urban area and we present some examples of this. The spatial mapping examples highlight the use of the SIFT-MS to reveal hotspots of pollutants, which can be investigated by further measurements in that area or through analysis which helps to reveal emissions sources. The correlations of species has been shown using a correlation matrix alongside hierarchical clustering, which groups species dependent on their similarities and can be used to determine emissions sources of certain species. Correlations between species

can be used to determine further analysis for the measurements made by the SIFT-MS in the mobile laboratory. For example, compound correlations can be further investigated through ratio analysis, which can be indicative of specific emissions sources. We have investigated the toluene to benzene ratio to determine the influence of different sources on their emissions. Gaussian kernel weighting was directly used in quantile regression modelling to provide an estimate of the T/B ratio spatially, which indicated evaporative emissions at higher quantile values. These methods will be further developed in the future for source

apportionment and characterisation of different source types along the whole of the York route, which can be used to increase understanding of air pollution in urban areas and for future emissions regulations.

   A UV photometric $O_3$ instrument will be added together with an ICAD $NO_x$ instrument, which will make for good comparison to the SIFT-MS measured $NO_2$. Future applications will be to return to measurements around York once travel activities return to normal, as the current COVID-19 situation appears to have had an effect on air pollution during the measurement

period due to decreased traffic and closed facilities. The upgrades discussed here highlight the potential to investigate vehicle

emissions as the use of the SIFT-MS allows for direct measurements of high concentration tailpipe emissions and for investigation of currently unregulated pollutants from vehicles. Future experiments investigating vehicle emissions include plume chase, where the mobile laboratory would follow a vehicle and sample the exhaust emissions for a period of time, and individual vehicle exhaust plume sampling, using the laboratory alongside remote sensing instruments by the side of the road and sampling vehicles as they pass. Additionally, direct measurements from the tailpipe will be used to highlight compounds of interest that can then be targeted by the SIFT-MS.

## Appendix A

| Date | No. of routes | Time of routes |
|------|---------------|----------------|
| **30th June 2020** | 2 | 13:03–13:42; 13:54–14:37 |
| **1st July 2020** | 3 | 11:18–11:57; 12:47–13:33; 13:58–14:40 |
| **2nd July 2020** | 3 | 13:20–14:07; 14:17–15:02; 15:35–16:13 |
| **3rd July 2020** | 4 | 10:58–11:44; 12:05–12:52; 13:29–14:21; 14:35–15:20 |
| **6th July 2020** | 3 | 11:40–12:19; 12:41–13:22; 13:42–14:25 |
| **8th July 2020** | 3 | 11:51–12:35; 13:21–14:07; 14:17–14:59 |
| **9th July 2020** | 4 | 11:26–12:10; 12:28–13:10; 13:38–14:25; 14:50–15:40 |
| **20th July 2020** | 3 | 12:00–12:48; 13:00–13:44; 14:32–15:13 |
| **21st July 2020** | 3 | 11:12–12:41; 12:42–13:30; 13:51–14:45 |
| **23rd July 2020** | 2 | 11:02–11:47; 12:28–13:17 |

**Table A1.** Details of the number of times the route was driven on each day and the times that the drives took place.

| Species | MM[1] |
|---|---|
| Ethylene | 28 |
| Isobutane | 58 |
| Benzene | 78 |
| Toluene | 92 |
| Ethylbenzene | 106 |
| Tetrafluorobenzene | 150 |
| Hexafluorobenzene | 186 |
| Octafluorotoluene | 236 |

[1]Molar Mass in $\mathrm{g\,mol^{-1}}$

**Table B1.** Compounds and their molecular masses in the Syft 2ppm standard gas, which was used to generate the daily instrument calibration factor (ICF).

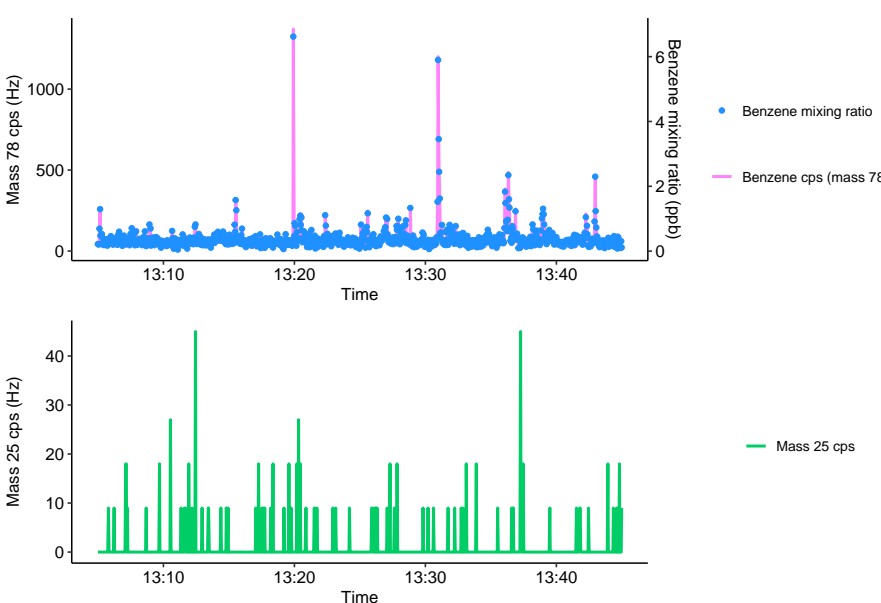

**Figure A1.** Example of instrument noise during mobile measurements. The blue points show the mixing ratio of benzene and the pink line shows the corresponding counts per second for the benzene ion counts per second (mass 78). The green line shows the variation of mass 25 during mobile measurements, which does not contain any periods of elevated noise.

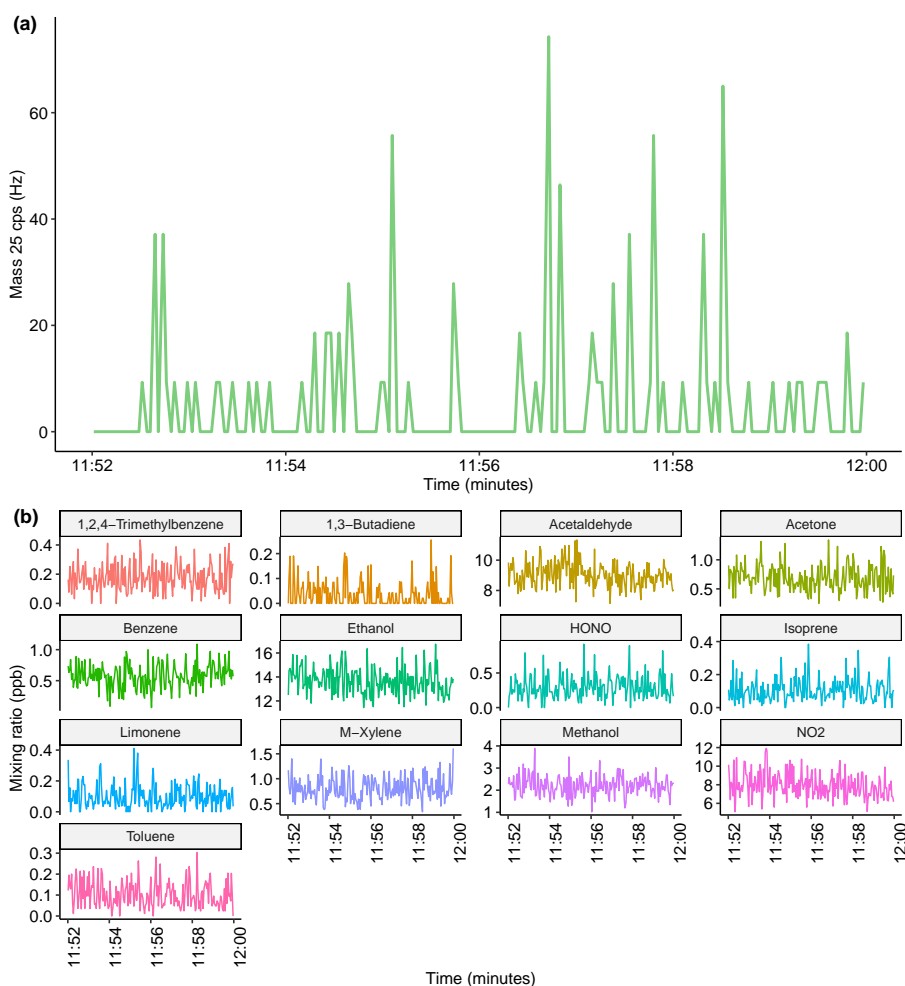

**Figure B1.** (a) Counts per second of mass 25 which represents instrument noise and (b) mixing ratios of the compounds during a nitrogen-only mobile measurement.

*Author contributions.* RW and MS designed and carried out the measurements. AC, JH and SY designed and built the mobile laboratory. RW prepared the manuscript with contributions from all co-authors.

*Competing interests.* The authors declare they have no conflict of interest.

*Acknowledgements.* The financial support from the NERC Panorama Doctoral Training Partnership for Rebecca Wagner is gratefully acknowledged. The authors would also like to thank Stuart Grange and Shona Wilde for help with data analysis and visualisation. We would also like to thank Chris Anthony for driving of the WASP around York.

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
