# Peer review of "Application of a mobile laboratory using a Selected-Ion Flow-Tube Mass Spectrometer (SIFT-MS) for characterisation of volatile organic compounds and atmospheric trace gases"

_Atmospheric Measurement Techniques, 2021_

## Author Comment (AC1)

**Response to the Reviewer's Comments**

We thank the reviewer for their comments and below is a response to the reviewer's general comments, which addresses the major points that they have highlighted. We have also provided responses to the specific comments and highlighted changes in the manuscript.

We have now included a comparison of SIFT-MS to PTR-MS in the introduction and highlighted SIFT-MS as an easier to use instrument that could be used by non-research organisations. Importantly SIFT-MS benefits from the use of three reagent ions, which can be switched in real-time, therefore allowing for measurements of a greater variety of species and separation of complex mixtures in atmospheric air. We thank the reviewer for highlighting mobile measurement studies carried out using a PTR-TOF-MS and we have now included these in our introduction.

We thank the reviewer for suggestions of scientific analysis that we could include in the paper. In the paper we have now expanded on the results and analysis section by developing a new method that is complementary to the existing analysis in the paper. This analysis is now in the paper as an additional section (Section 3.4) and involves investigating the toluene to benzene (T/B) ratio using quantile regression modelling. This has been used alongside a gaussian kernel smoother to show the T/B ratio along the distance of a road of interest to reveal intermittent sources that would be missed through only spatially averaging compounds.

We hope that these responses are satisfactory and that incorporation of a further discussion of the differences of SIFT-MS compared to PTR-MS and further development of the results section has addressed the points raised by the reviewer.

**Reviewer 1**

The authors present measurements using a mobile laboratory equipped with a Selected-Ion Flow-Tube Mass Spectrometer (SIFT-MS) to measure 13 volatile organic compounds (VOCs) including aromatics, monoterpenes, nitrogen oxide, and more abundant species like methanol, ethanol, and acetone together with other instrumentation to measure $CO_2$ and $CH_4$. They provide details on the mobile laboratory setup including the online calibration of compounds. A set of measurements is presented where they perform a correlation analysis of all compounds and hierarchical clustering.

My main concern is that this study is not presenting a new method that isn't already published by previous work. Although the SIFT-MS is a great addition to the mobile lab such measurements have been performed before by higher-resolution instruments. An example is the proton transfer reaction time of flight mass spectrometer (PTR-ToF-MS) measuring hundreds of VOCs at 1-sec resolution that has been extensively used by the NOAA team but not referenced at all in the introduction. Recent example publications are from Coggon et al. (2016); Yuan et al. (2017); Coggon et al. (2018); Shah et al. (2020); Gkatzelis et al. (2021a); Gkatzelis et al. (2021b); Stockwell et al. (2020). TOFWERK has also been using the VOCUS for real-time measurements on their mobile lab in Europe (`https://www.tofwerk.com/vocus-ptr-mobile-laboratory-video/`), Aerodyne in the US e.g. for profiling natural gas production (`https://www.tofwerk.com/oil-and-gas-well-emissions/`), and the same type of measurements have been performed by Montrose (`https://montrose-env.com/services/testing-lab-services/ptr-tof-ms-mobile-laboratory/`), and the RJ Lee group (e.g. `https://rjlg.com/2019/06/philadelphia-refinery-explosion-air-quality-impact/`). Furthermore, a characteristic paper similarly discussing this method was published last year by Richards et al. (2020) on mobile measurements of VOCs using a PTR-ToF-MS and a membrane introduction mass spectrometry (MIMS) where they also perform more detailed Principal Component Analysis (PCA) to source apportion VOCs. This literature can be easily found by just googling "mobile laboratory measurements of VOCs" so I am surprised it is currently not covered by the authors. I am therefore not convinced that this is truly a new method. I could see how this work could be published as a complementary, possibly cheaper (?) way to perform such measurements. Of course, I leave this to the editor to decide. Nevertheless, carefully promoting all the existing work and the benefits and advantages of other studies compared to this one will be crucial, something the paper is currently lacking.

Regarding the scientific analysis of the derived data I find it to be limited to just correlations that are not fully discussed. A benefit I see from the SIFT-MS is that it can measure VOCs but also $NO_2$. The slopes of the correlations of VOCs to $NO_2$ could provide more insights into the pollution source. This correlation analysis would also be more informative when performed at higher resolution as a rolling

correlation function. Analysis to prove whether the measurements obtained here are influenced by traffic should focus on comparisons to previous literature. What are the obtained slopes from the correlations for every few minutes of data and how do they compare to a vast literature of traffic emissions? These correlations could be done both for VOCs vs. NO2 but also VOCs vs. CO2. Also, I would expect enhancements of emission when moving to the city center. Did the authors observe that as has been seen before by various other groups (e.g. `https://www.fz-juelich.de/iek/iek-8/EN/Expertise/Infrastructure/MobiLab/MobiLab_node.html`)? Some timeseries plots or enhancement box-and-whiskers would be great additions here. Furthermore, the authors at points of the manuscript discuss the influence of other sources like paints and in general volatile chemical products. It would be great to compare their results to inventory estimates by McDonald et al. (2018) and paint studies e.g. Stockwell et al. (2020). While this is a measurement technique paper I do consider that science that validates the importance of performing such measurements should be included and in my opinion is currently not sufficient. Statistical methods to separate to different sources would also be valuable e.g. the use of positive matrix factorization. If this is not an option then I would consider discussing in detail the benefits of performing such statistical approaches in the future.

Overall, rewriting the introduction, focusing on what is new from these measurements, discussing the benefits compared to other published work as well as the disadvantages, and performing further analysis of the obtained data would be the main improvements before this publication is suited for AMT. Below some additional specific comments.

1.1 *Acetone, methanol, and ethanol are abundant species coming from multiple sources including atmospheric chemistry. This may drive the correlations observed here. Slopes would be interesting to have and compare to other studies. More detailed timeseries of compounds for specific events would be helpful to further conclude on possible sources.*

    Since publishing the paper, we have applied a compound specific zero offset to the measurements for each individual day. When re-plotting the correlation plot, the correlations between these species was not as strong as it was previously, so no further analysis has been carried out into these species.

1.2 *Table 4 should be a box and whiskers figure with the y-axis representing the compound names and the x-axis the concentrations. In-city and out-of-city box and whiskers would be valuable and provide urban enhancements during the COVID-19 pandemic.*

    Table 4 has now been changed into a box and whisker plot.

1.3 *Also, a more detailed discussion on the lockdown conditions during the period of the measurements would be great. A suggested reference could be the stringency index $https://ourworldindata.org/grapher/covid-stringency-index$).*

    We thank the author for the suggested reference and a sentence has been added which comments on the lockdown conditions.

    " It should be noted that during the measurement period the Covid-19 stringency index was 64.35 (taken from: `https://ourworldindata.org/grapher/covid-stringency-index?tab=chart&country=~GBR`), indicating significantly reduced economic and traffic activity. This could affect both the concentration and detection of different VOC species due to possible decreased emissions. "

1.4 Line 10: *Correct to "sources".*

    Suggestion implemented.

1.5 Lines 22-32: *The authors could further discuss here VOC emissions from other pollution sources in more detail. Volatile chemical products, cooking emissions, residential wood burning, and industry with their respective citations would be of value here. Also, studies that focus on the contribution of different sectors of VOC emissions would be valuable too.*

    A discussion of the contribution to VOC emissions from different emissions sources has been added to the introduction.

> "Mass balance analysis by McDonald et al. (2018) concluded that emissions from VCPs now account for half of the fossil fuel VOC emissions in industrialised cities. Lewis et al. (2020) estimated that in the UK, VOC emissions from solvent use and industrial processes account for 63% of total VOC emissions."

1.6 Line 29: *Add references. Examples for volatile chemical products could be McDonald et al. (2018), Stockwell et al. (2020), Gkatzelis et al. (2021a,b).*

We have added the reference from McDonald et al. (2018) (as shown above) when discussing the contribution to VOC emissions. We have also added additional VOC sources in the text, but did not feel including references for all of the different emissions sources was necessary. We have instead included these references further along on the introduction discussing the use of PTR-MS in mobile laboratories to identify emissions sources.

> "Nevertheless, the reduction in vehicular emissions means it is likely that other sources of emissions, such as volatile chemical products (VCPs), solvent use, cooking, residential wood burning and industry, have become more important."

1.7 Lines 57-58: *Many recent studies use PTR-ToF-MS in mobile laboratories that are currently not discussed. See comments above.*

This section has been re-written to include studies that use PTR-ToF-MS in mobile laboratories. We thank the reviewer for a comprehensive list of references.

1.8 Line 123: *Correct to "formulas".*

Suggestion implemented.

1.9 Line 161: *delete "instrument"*

Suggestion implemented.

**References**

Matthew M. Coggon, Patrick R. Veres, Bin Yuan, Abigail Koss, Carsten Warneke, Jessica B. Gilman, Brian M. Lerner, Jeff Peischl, Kenneth C. Aikin, Chelsea E. Stockwell, Lindsay E. Hatch, Thomas B. Ryerson, James M. Roberts, Robert J. Yokelson, and Joost A. de Gouw. Emissions of nitrogen-containing organic compounds from the burning of herbaceous and arboraceous biomass: Fuel composition dependence and the variability of commonly used nitrile tracers. *Geophysical Research Letters*, 43(18):9903–9912, 9 2016. ISSN 00948276. doi: 10.1002/2016GL070562. URL http://doi.wiley.com/10.1002/2016GL070562.

Bin Yuan, Matthew M. Coggon, Abigail R. Koss, Carsten Warneke, Scott Eilerman, Jeff Peischl, Kenneth C. Aikin, Thomas B. Ryerson, and Joost A. De Gouw. Emissions of volatile organic compounds (VOCs) from concentrated animal feeding operations (CAFOs): Chemical compositions and separation of sources. *Atmospheric Chemistry and Physics*, 17(8):4945–4956, 4 2017. ISSN 16807324. doi: 10.5194/acp-17-4945-2017.

Matthew M. Coggon, Brian C. McDonald, Alexander Vlasenko, Patrick R. Veres, François Bernard, Abigail R. Koss, Bin Yuan, Jessica B. Gilman, Jeff Peischl, Kenneth C. Aikin, Justin Durant, Carsten Warneke, Shao Meng Li, and Joost A. De Gouw. Diurnal Variability and Emission Pattern of Decamethylcyclopentasiloxane (D5) from the Application of Personal Care Products in Two North American Cities. *Environmental Science and Technology*, 52(10):5610–5618, 5 2018. ISSN 15205851. doi: 10.1021/acs.est.8b00506. URL https://pubs.acs.org/sharingguidelines.

Rishabh U. Shah, Matthew M. Coggon, Georgios I. Gkatzelis, Brian C. McDonald, Antonios Tasoglou, Heinz Huber, Jessica Gilman, Carsten Warneke, Allen L. Robinson, and Albert A. Presto. Urban Oxidation Flow Reactor Measurements Reveal Significant Secondary Organic Aerosol Contributions from Volatile Emissions of Emerging Importance. *Environmental Science and Technology*, 54(2): 714–725, 1 2020. ISSN 15205851. doi: 10.1021/acs.est.9b06531. URL https://pubs.acs.org/sharingguidelines.

Georgios I. Gkatzelis, Matthew M. Coggon, Brian C. McDonald, Jeff Peischl, Kenneth C. Aikin, Jessica B. Gilman, Michael Trainer, and Carsten Warneke. Identifying Volatile Chemical Product Tracer Compounds in U.S. Cities. *Environmental Science and Technology*, 55(1):188–199, 1 2021a. ISSN 15205851. doi: 10.1021/acs.est.0c05467. URL `https://dx.doi.org/10.1021/acs.est.0c05467`.

Georgios I. Gkatzelis, Matthew M. Coggon, Brian C. Mcdonald, Jeff Peischl, Jessica B. Gilman, Kenneth C. Aikin, Michael A. Robinson, Francesco Canonaco, Andre S.H. Prevot, Michael Trainer, and Carsten Warneke. Observations Confirm that Volatile Chemical Products Are a Major Source of Petrochemical Emissions in U.S. Cities. *Environmental Science and Technology*, 55:4343, 2021b. ISSN 15205851. doi: 10.1021/acs.est.0c05471. URL `https://dx.doi.org/10.1021/acs.est.0c05471`.

Chelsea Stockwell, Matthew Coggon, Georgios Gkatzelis, John Ortega, Brian McDonald, Jeff Peischl, Kenneth Aikin, Jessica Gilman, Michael Trainer, and Carsten Warneke. Volatile organic compound emissions from solvent- and water-borne coatings: compositional differences and tracer compound identifications. *Atmospheric Chemistry and Physics Discussions*, pages 1–29, 2020. ISSN 1680-7316. doi: 10.5194/acp-2020-1078.

L. C. Richards, N. G. Davey, C. G. Gill, and E. T. Krogh. Discrimination and geo-spatial mapping of atmospheric VOC sources using full scan direct mass spectral data collected from a moving vehicle. *Environmental Science: Processes and Impacts*, 22(1):173–186, 1 2020. ISSN 20507895. doi: 10.1039/c9em00439d. URL `https://pubs.rsc.org/en/content/articlehtml/2020/em/c9em00439dhttps://pubs.rsc.org/en/content/articlelanding/2020/em/c9em00439d`.

Brian C. McDonald, Joost A. De Gouw, Jessica B. Gilman, Shantanu H. Jathar, Ali Akherati, Christopher D. Cappa, Jose L. Jimenez, Julia Lee-Taylor, Patrick L. Hayes, Stuart A. McKeen, Yu Yan Cui, Si Wan Kim, Drew R. Gentner, Gabriel Isaacman-VanWertz, Allen H. Goldstein, Robert A. Harley, Gregory J. Frost, James M. Roberts, Thomas B. Ryerson, and Michael Trainer. Volatile chemical products emerging as largest petrochemical source of urban organic emissions. *Science*, 359(6377):760–764, 2 2018. ISSN 10959203. doi: 10.1126/science.aaq0524. URL `http://science.sciencemag.org/`.

Alastair C. Lewis, Jim R. Hopkins, David C. Carslaw, Jacqueline F. Hamilton, Beth S. Nelson, Gareth Stewart, James Dernie, Neil Passant, and Tim Murrells. An increasing role for solvent emissions and implications for future measurements of volatile organic compounds: Solvent emissions of VOCs. *Philosophical Transactions of the Royal Society A: Mathematical, Physical and Engineering Sciences*, 378(2183), 10 2020. ISSN 1364503X. doi: 10.1098/rsta.2019.0328.

---

## Author Comment (AC2)

**Response to the Reviewer's Comments**

We thank the reviewer for their comments and below is a response to the reviewer's general comments, which addresses the major points that they have highlighted. We have also provided responses to the specific comments and highlighted changes in the manuscript.

The time delay is discussed in the specific comments section (lines 82-84) and this was tested using a lighter, a 10 second time delay for the UGGA was applied and no time delay was observed for the SIFT-MS. Wind conditions were not investigated as the number of routes that were performed should cover a range of meteorological conditions as they were performed over multiple days.

Further discussion on the calibration approach has been added to provide clarity on the approach and the reasoning behind it. A detailed response is included below (lines 154-158).

In the paper we have now expanded on the results and analysis section by developing a new method that is complementary to the existing analysis in the paper. This analysis is now in the paper as an additional section (Section 3.4) and involves investigating the toluene to benzene (T/B) ratio using quantile regression modelling. This has been used alongside a gaussian kernel smoother to show the T/B ratio along the distance of a road of interest to reveal intermittent sources that would be missed through only spatially averaging compounds.

Further analysis will be carried out to continue the characterisation of different emission sources in York and further development of the new method of quantile regression modelling will be carried out, which will also assist with determination of dominant emission sources. $C_2$-alkylbenzenes and $C_3$-alkylbenzenes may not all be required to identify gasoline emissions, but both of these classes of compounds were in the calibration gas, so it was useful to measure these externally calibrated compounds to determine the success of the new measurement technique. It is worth noting that the selected compounds can be changed to suit different applications, so this will be done for future measurements. Ratios of toluene-to-benzene are investigated in the results section and are compared to other studies to help distinguish different emissions sources. The results section discusses the use of toluene-to-benzene to distinguish between emissions and evaporative/solvent emissions along certain sections in York. Other sources such as hairdressers and dry cleaners were not revealed by spatial mapping. However with further development of the quantile regression analysis, it is likely that other intermittent sources will be revealed.

We hope that these responses are satisfactory and that incorporation of a further discussion of the calibration approach and further development of the results section has addressed the points raised by the reviewer.

**Reviewer 2**

This study presents the development of a mobile analytical platform equipped with two analyzers: an Ultraportable Greenhouse Gas Analyzer (UGGA) and a Selected-Ion Flow-Tube Mass Spectrometer (SIFT-MS) measuring carbon dioxide, methane, several VOCs and other trace gases. The authors describe the different techniques used by these instruments and give some details about the calibration of the SIFT-MS. This mobile platform was deployed in the city of York, UK, where it completed a total of 31 surveys of the same route over a 10-day period. The authors present here some preliminary results of these measurements.

Overall, this study has a lot of potential and could be a valuable contribution to the literature as it is important to monitor pollutant emissions and their evolution in urban areas. Mobile measurements of VOCS are very useful especially to characterize and identify different types of sources within a city. However, I was expecting a deeper analysis of the measurements. The development of such a mobile platform represents a lot of work but it is enough to justify a paper. The authors managed to collect an impressive amount of data (especially during a pandemic period), the paper would really benefit from developing the results section, I recommend addressing the following major points before publishing it:

It is not specified in the article if the authors took into account the time delay between the sampling inlet at the front of the van and the moment it reaches the instruments. This time delay can induce a shift of the measurements due to the motion of the vehicle and impact the aggregation of the measurements into the 30 metres segment. It is important to give this information when presenting mobile measurements. Did the authors look at the wind conditions during the measurements? If the wind

was coming from one direction on given day and on the opposite direction on another day, it will likely have an impact on the composition of the sampled air, especially when measurements are performed close to known sources.

I would have liked more discussions about the calibration approach. First, I would make it clear at the beginning of Section 3.1 that 2 or 3 different calibration approaches were used for different components (AGCU and equation 1 plus calibration of UGGA?). Second, I would explain why the authors chose this seven steps approach? I understand that due to time constraint, it is not possible to measure each step for too long but 3 minutes seem a bit short. Why not measure each step only once but twice as long? Why did the authors only used the right-hand steps of the post-drive calibration? What is the point of the pre-drive calibration? Did the authors observe any drift of the measurements between the first and the last day of the campaign? Third, I would have also been interested in a comparison of the two calibration methods. The authors could have applied both approaches to benzene measurements for example, and shown how the two sets of calibrated measurements compare.

Reorganize Section 3.2: having only one subsection does not make much sense. I would either add a title to the first part of Section 3.2 or remove "3.2.1 Spatial correlation mapping". Also, I do not understand why the authors presented the distribution maps of only 2 components, I would have liked to see more, especially species that are correlated (benzene, toluene, alkylbenzenes...). I was a bit disappointed at the end of this section when the authors teased the application of the correlation approach to smaller scales for a future study, I was expecting to see that in this paper... In the end, I am not sure I am convinced of the benefit of measuring all of these components at the same time. Here are a list of questions I got after reading this paper: What is the dominant source of VOCs of York? Do we really need C2-alkylbenzenes and C3-alkylbenzenes measurements to identify emissions from gasoline evaporation? What are the ratios of these different components for known sources? How do they compare with ratios found in other studies for the same type of sources? Can we differentiate emissions from gasoline fuel and diesel with one of the measured component? What about the other sources mentioned in the paper (dry cleaners, hairdressers...), were you able to detect any emissions from them?

1.1 Line 1: *"The importance of emissions source types...": the authors should specify what type of sources they are referring to (VOCs? GHG? atmospheric pollutants?).*

This sentence has been changed accordingly.

"Over the last two decades, the importance of emissions source types of atmospheric pollutants..."

1.2 Lines 23-24: *Be consistent with the notation throughout the text: choose between "ozone" and "O3" and stick to it. Same for "secondary organic aerosol"/SOA, "limit of detection"/LOD...*

Suggestion implemented.

1.3 Lines 29-30: *I would specify that you are talking about static measurements at monitoring sites here. This is developed later but it is not very clear in this sentence*

Suggestion implemented.

1.4 Line 50: *Replace "targeted" by "targeting"?*

This sentence has now been removed.

1.5 Line 83: *"1/2 inch": shouldn't the authors use SI units?*

SI units have now been included.

1.6 Lines 82-84: *How long does it take for the sampled air to go from the inlet in front of the car to the instrument? Is this delay taken into account and corrected on your maps? If not, it will have an impact when you bin your data into 30 meters segments.*

The time delay for the instruments was investigated by using a lighter at the front of the sample line and observing the time delay in an increase in butane (SIFT-MS) and $CO_2$ (UGGA). The SIFT-MS gave an instant response to the lighter test so no offset had to be applied. The time delay of the UGGA was 10 seconds so this offset was applied to the measurements of the UGGA. Since the measurements, the pump of the UGGA has been changed, so any future measurement will have a faster response.

1.7 Line 86: *Is the wind corrected for the motion of the platform?*

The wind was not corrected for the motion of the platform and therefore was not used in the study. The text now reads:

"Real-time location of the WASP is recorded by a Garmin GPS 18x PC and a measurement of wind speed and direction is measured using a Gill 2D Ultrasonic Wind Sensor (which was not used in this study)..."

1.8 Section 2.2: *The authors give the precision of the UGGA for methane and carbon dioxide but they do not talk about the precision of the SIFT-MS. What is the precision of the SIFT-MS for the different species? Does it correspond to the limit of detection described in Section 3? What is the limit of detection of the UGGA for methane and carbon dioxide?*

The precision of the SIFT-MS measurements has been added to Table 3 and was calculated using 2 times the standard deviation ($\sigma$) of measurements made when sampling from zero air or nitrogen gas.

1.9 Line 120: *How many species does the SIFT-MS actually measures? This is very confusing: Table 2 and Section 2.2.1 state that it measures 13 components but Table 1 states 14 species.*

The SIFT-MS measures 13 during the measurement period, there was a mistake in table 1 and we thank the reviewer for noticing this.

1.10 Lines 123-125: *I am not sure I correctly understand what is a cycle? Does it correspond to the measurement with one reagent or to the succession of the three reagents?*

To avoid confusion to the reader, the term cycle time has been removed and acquisition rate added. The text now reads:

" To maximise spatial data density during mobile measurements,the instrument acquisition rate was minimised with only a single product ion monitored for each compound. Therefore the sampling method used during measurements has an acquisition rate of 2.5 seconds with a 90 ms ion dwell time."

1.11 Lines 134-135: *The precision of an instrument should always be stated over a time period.*

The precision time stated in the text is over 1 second and the time period has been included in the text.

1.12 Lines 154-158: *Why do you only use the right-hand steps of the post-drive for the calibration? Did you test several calibration approach with the AGCU? I wonder why you implemented these seven consecutive steps of three minutes instead of only doing four steps but with longer measurements. Also, each step of the calibration last 3 minutes, did you use all the data within the 3 minutes or did you get rid of the first measurements that are usually influenced by the transition? Why is the agreement between pre- and post-drive calibration not as good for ethanol and methanol?*

The calibration set up consisted of a heated palladium catalyst (380°C) to generate clean zero diluent gas from ambient air coupled with a gas blender to deliver controlled gas concentrations to the SIFT-MS. As this calibration set up had to be switched off at the end of each day and restarted the following morning (for safety and practical reasons) there had to be adequate time for the catalyst and calibration system internal surfaces to condition with the target VOCs on a daily basis. This is why only the right-hand steps of the post drive calibration were used. Practically these produced the most agreeable results.

We tested several calibration profiles, but the main problem is the time constraints involved. In order to get the SIFT-MS operational (which has to be switched on and off each day), stable and calibrated pre and post drive and then spend several hours performing mobile measurements on a daily basis there are considerable time constraints, hence the calibration protocol has to be concise. 4 calibration points were chosen as this is considered to be the minimum requirement for accurate instrument calibration. 2 x 3 minute calibration steps were chosen to assess both the reproducibility within the calibration and across pre- and post-drive calibrations. In our opinion this is a minimum requirement for confidence in any calibration.

We did not use the first two minutes of the calibration, this was regarded as an "equilibration period" at that step. The average of the last minute of each step was used for the applied calibration. The poorer agreement between the pre and post calibrations for methanol and ethanol is unclear. However, in our experience both methanol and ethanol residence times in PFA sample tubing are influenced by the humidity of the gas. We believe that the zero air generator (which provides VOC free air from ambient air whilst maintaining its humidity) has had insufficient time to warm up before the pre-drive calibrations. Additional text has been added to the calibration discussion to explain this.

1.13 Line 161: *Delete "instrument" in "The daily instrument ICF was derived...".*

Suggestion implemented.

1.14 Line 162: *Add a space between "2" and "ppm".*

Suggestion implemented.

1.15 Lines 175-178: *It would have probably be more intuitive to test the effect of the movement and vibration of the van by comparing measurements when the platform is in motion and when it is parked.*

We feel that testing the effect of the movement and vibration of the van on the SIFT-MS whilst driving is a sufficient test as this gave us the same conditions as when the mobile measurements were carried out.

1.16 Line 178: *Why is there no mention of the calibration procedure for the UGGA? Did you also calibrated this analyzer before and after each drive with a multi-point calibration approach? This should be discussed here.*

A quantitative measurement of mixing ratio could be determined directly from the UGGA as the measured absorption spectra is recorded and combined with measured gas temperature and pressure in the cell, effective path length and known line strength. Therefore, the UGGA could be used without performing external calibration for each drive. The UGGA was externally calibrated before and after the measurement period using external gas cylinders. A sentence has been added to the text.

"A quantitative measurement of mixing ratio could be determined directly from the UGGA without the need for calibration during each drive, but the UGGA was calibrated with external gas cylinders before and after the measurement period."

1.17 Lines 188-190: *I do not understand this sentence: why does having more data points in bins than in your limit of detection test confirms the confidence of the SIFT-MS measurements?*

This sentence has been removed from the text.

1.18 Figures 1 and 2: *I would combine these two figures.*

We think that the figures work better separately in their respective sections for the WASP and the SIFT-MS.

1.19 Figure 6: *Change the legend of the benzene plot to better emphasize that the cps are represented by the blue dots (and only the dots) and the mixing ratio are represented by the pink line (without dots).*

Suggestion implemented.

**1.20 Figure 8:** *Replace" No2" and "Hono" by "NO2" and "HONO", respectively.*

Suggestion implemented

**1.21 Figure A1:** *Replace "No2" by "NO2".*

Suggestion implemented and "Hono" has been changed to "HONO" as suggested for Figure 8.

**1.22 Table 1:** *The second column most likely indicates the measurements frequency or resolution rather than the response time of the instrument.*

The header on the second column has now been changed to time resolution.

---

## Author Response (AR2)

**1 Response to the Reviewer's Comments**

We thank the reviewer for their comments and below is a response to the reviewer's specific and technical comments and highlighted changes in the manuscript. We hope that these responses are satisfactory and that polishing the introduction by removing any repetition and adding further text to captions/discussion of the figures has addressed the points raised by the reviewer.

**Report 1**

The authors improved the manuscript by 1. providing a more detailed introduction, 2. describing in more detail the calibration setup, and 3. extending the scientific analysis by looking into the benzene to toluene ratio. The introduction needs to be polished since the authors often repeat sentences. Regarding the new scientific analysis, it will be great if the authors clearly mention which datasets/drives are used for each of the graphs, specifically for Figure 8- Figure 11.

**Specific comments**

1.1 Line 18-19: *PAN is also known to thermally decompose and form NOx that can affect O3 production.*

Some text has been added that states this.

"and it has also been shown to thermally decompose to form $NO_x$, which leads to enhanced $O_3$ production (Heald et al., 2003)"

1.2 Line 20-21: *You are repeating that VOCs have health impacts...*

This sentence has now been changed

"Also, some VOCs can cause..."

1.3 Line 84-86: *Repeating what is already mentioned*

Any repetitions have now been removed.

1.4 Line 224: *Are these box-and whiskers generated for all 30 drives? Please specify.*

Yes the box and whisker plots are generated from the data containing all of the 30 drives and the caption now states this.

"Summary of measurements made by a) the SIFT-MS (in ppb) and b) the UGGA (in PPM) during 30 repeat drives around York."

1.5 Line 226-227: *What about isoprene?*

The measurement route around York does not contain any large urban green spaces and therefore we expect the mixing ratios to be relatively low. We agree with our statement that there is a 'lack of emission sources for these compounds in York'.

1.6 Line 230-231: *Ethanol could also originate from VCPs.*

A sentence has now been added stating this.

"Ethanol variation could also result from use of VCPs."

1.7 Line 254: *Is this the most populated area of the town? Some discussion on the population density may be of interest here since it may indicate human sources.*

This area of the centre would not be expected to be the most populated area and it is dominated by businesses and shops. It is likely therefore that increases here are due to commercial premises.

1.8 Line 264-274: *Please discuss the expected high background concentrations of the non-vehicle related compounds. Could this play a role in their grouping performed?*

It is difficult to say whether the grouping is due to high background concentrations or similar emission sources. But, it is worth noting that the correlation coefficients are quite low ($> 0.5$).

1.9 Line 282-284: *Would you expect emissions of benzene and toluene from other sources? What about cooking? Biomass burning? Are these sources expected during the period of the measurements?*

We would not expect biomass burning to be an important emission of benzene and toluene as the measurements were made during the Summertime, so burning in homes would not be taking place. Toluene and benzene correlate the best to tracers from gasoline fuel evaporation or engine exhaust emissions (as shown in the correlation matrix), so cooking is not expected to be an important source.

1.10 Figure 6: *You could add the LOD as a marker per compound in the graph.*

We feel that keeping Figure 6 as presented is preferable as adding a marker for LoD may make the plot less clear for the reader.

1.11 Figure 8: *Provide more explanations in the caption. What are the numbers and colors indicating? Also, please indicate what the lines on the right are in more detail.*

Suggestion implemented.

"A higher correlation coefficient between species is represented by a higher number, a darker red colour and an ellipses (shape). The lines on the right-hand side show the hierarchical clustering between compounds and represent clusters of species with similar patterns/behaviours."

**Technical comments**

1.12 Line 177: *Delete "significantly" since the stringency index is an indicator but doesn't directly describe the pollution reduction.*

Suggestion implemented.

1.13 Line 253: *Replace ", this" to " that"*

This sentence has now been changed.

1.14 Line 266: *Change to "evaporation or engine exhaust"*

Suggestion implemented.

1.15 Line 252-253: *I would rephrase to "Ethanol is at background levels. . . "*

Suggestion implemented.

1.16 Line 280: *Repeating "many studies".*

We have now removed 'many studies' from the previous sentence.

**2 Response to the Reviewer's Comments**

We thank the reviewer for their comments and below is a response to the reviewer's general comments, which addresses the points that they have highlighted. We have also provided responses to the specific comments and highlighted changes in the manuscript. We hope that these responses are satisfactory and that reworking section 3.4 and adding further text to captions/discussion of the figures has addressed the points raised by the reviewer.

**Report 2**

Review of "Application of a mobile laboratory using a Selected-Ion Flow-Tube Mass Spectrometer (SIFT-MS) for characterization of volatile organic compounds and atmospheric trace gases" by Wagner et al., for publication in AMT

This study presents the application of a mobile analytical platform equipped with two analyzers: an Ultraportable Greenhouse Gas Analyzer (UGGA) and a Selected-Ion Flow-Tube Mass Spectrometer (SIFT-MS) measuring carbon dioxide, methane, several VOCs and other trace gases. This mobile platform was deployed in the city of York, UK, where it completed a total of 31 surveys of the same route over a 10-day period.

Upon re-reviewing the article, I really appreciate that the authors made a thorough effort to address the several comments received during the first review process. The authors had particular care in further analyzing their measurements and developing the results section. Indeed, they added a section where they investigated the toluene to benzene ratio of their measurements to differentiate between emissions from vehicle exhausts and evaporative emissions from fuels and solvents. Even if I find this added section a bit confusing (see comments below) and think it should be reworked a bit, I am satisfied with the other edits and responses to most review comments. Overall, this manuscript should be published subject to these minor corrections:

1.17 : *There are many acronyms in this article (especially in the first two sections), some of them are defined once but never used after (CAFO) or only used once (PAN, ARI), some are defined several times (UGGA), some are not defined at all (PTR-TOF-MS). I would recommend simplifying all of these notations by getting rid of the ones that are almost never used. For PTR-TOF-MS, I would describe this technique, explain the difference with regular PTR-MS and indicate right away that it will be referred as PTR-TOF in the rest of the article.*

> We have simplified and removed notations throughout the text. We have also added some text describing PTR-MS and PTR-TOF-MS and the differences.

> "A PTR-MS is a term used for an instrument which consists of an ion source that is directly connected to a drift tube and a mass analyzing system, which either consists of a quadrupole of time of flight mass analyser. Standard PTR-MS instruments are a PTR-QMS (Proton Transfer Reaction Quadrupole Mass Spectrometer), which can detect and resolve product ion masses at single unit mass resolution. ... A PTR-TOF can detect and resolve product ions at much higher mass resolution with currently available commercial instruments having a mass resolution ¿ 4000."

1.18 *I would be consistent with the introduction of the different species' chemical formulas: some species have their chemical formula introduced properly (O3), other formulas are used in the introduction and introduced in section 2 (CH4, CO2), other formulas are simply never introduced (NO2, HONO, NH3...).*

> Suggestion implemented.

1.19 *Section 3.4 is a bit confusing to me. In this section, the authors describe how the T/B ratio can be used to differentiate emissions from vehicle exhausts and evaporative emissions from fuels and solvents, and explain how both of these sources can be transient. I am not sure I understand what do the authors mean L294-296: "Taking the median value of concentrations will down-weight the contribution from less frequent, higher concentrations as used by Apte et al. (2017). However, there is potentially important information that can be missed when only considering the median",*

*are they talking about using the median value of toluene and benzene in each 30 metre segment of road to calculate the T/B ratios rather than one of the regressions described later? Or using an ordinary least square regression? I think the authors should guide the reader by introducing the 2 possible approaches to estimate the T/B ratio (OLS and quantile regression), and then explain their advantages and drawbacks (transient emissions...). In the end, I am not sure the T/B ratio example on Hull Road is the best one to illustrate the advantages of the quantile regression over the OLS: in both case these ratios are over 2 which indicates that the main source of the emissions is from fuels and solvents. Overall, this section should be reworked a little bit to make it easier for the reader to follow.*

Line 294-296 is discussing the median ratio used for spatial mapping, which has now been stated in the text. The T/B ratio plot shown in Figure 9 shows the OLS regression slope obtained from the data from all 30 repeat drives around York, but each point represents a 30 metre segment of road. We have reworked this section but have kept the same structure, we hope that it now reads more clearly. We feel that Hull road is a good example to show the differences between OLS and quantile regression as there is a clear difference in the T/B ratio seen at higher percentiles compared with the 50th percentile. The two regression approaches may show some similarities, but the quantile regression used along a gaussian kernel smoother ensures that transient emissions are fully captured.

1.20 Figure 10: *I suggest to improve the title of this figure or at least remove "A plot of...".*

Suggestion implemented.

1.21 Figure 11: *I would add the slope values for the different regressions, I find it relatively difficult to compare "Evaporative" and "Remainder" as it is currently.*

Suggestion implemented.

1.22 Line 338: *I would remove "We have demonstrated the correlation method". The authors did observe correlations between species but the method does not need a demonstration.*

Suggestion implemented.

**References**

Colette L. Heald, Daniel J. Jacob, Arlene M. Fiore, Louisa K. Emmons, John C. Gille, Merritt N. Deeter, Juying Warner, David P. Edwards, James H. Crawford, Amy J. Hamlin, Glen W. Sachse, Edward V. Browell, Melody A. Avery, Stephanie A. Vay, David J. Westberg, Donald R. Blake, Hanwant B. Singh, Scott T. Sandholm, Robert W. Talbot, and Henry E. Fuelberg. Asian outflow and trans-Pacific transport of carbon monoxide and ozone pollution: An integrated satellite, aircraft, and model perspective. *Journal of Geophysical Research: Atmospheres*, 108(D24):4804, 12 2003. ISSN 2156-2202. doi: 10.1029/2003JD003507. URL https://agupubs.onlinelibrary.wiley.com/doi/full/10.1029/2003JD003507https://agupubs.onlinelibrary.wiley.com/doi/abs/10.1029/2003JD003507https://agupubs.onlinelibrary.wiley.com/doi/10.1029/2003JD003507.